# Quality of life of COVID 19 patients after discharge: Systematic review

H. M. R. K. G. Nandasena[1]*, M. L. Pathirathna[1], A. M. M. P. Atapattu[1], P. T. S. Prasanga[2]

1 Department of Nursing, Faculty of Allied Health Sciences, University of Peradeniya, Peradeniya, Sri Lanka.
2 Post Graduate Institute of Medicine, Faculty of Medicine, University of Colombo, Colombo, Sri Lanka

* renukalhari@ahs.pdn.ac.lk, renu88kalhari@gmail.com

## Abstract

### Background

Understanding the impact of COVID 19 on patients' quality of life (QOL) following discharge or recovery is essential for planning necessary interventions in advance. As such, this systematic review aimed to provide an overview of the QOL, and the factors associated with it in COVID-19 patients following discharge or recovery.

### Methods

The Databases of PubMed, Cochrane Library, and Science Direct were searched. The review included studies that (1) assessed the QOL of COVID 19 patients following discharge or recovery, (2) were written in English, (3) used a validated instrument to assess the quality of life and (4) used an observational or cohort study design. The PRISMA guidelines were followed. Following the initial search, 2866 articles were identified as being related. A total of 1089 articles were identified as duplicates. 1694 studies were excluded during the title and abstract screening stage, and 83 studies were screened at the full-text screening stage. Finally, 21 studies were included in this systematic review.

### Results

This systematic review included 4408 patients who tested positive for COVID 19. Of them 50.2% (n = 2212) were males. Regardless of the time since discharge or recovery, COVID 19 patients' QOL has been significantly impacted. Female sex, older age, co-morbidities, Intensive Care Unit (ICU) admission, prolonged ICU stay, and being mechanically ventilated were the most frequently reported factors associated with the low level of QOL.

### Conclusion

The QOL of the post COVID19 patients was significantly impacted, regardless of the time elapses since discharge or recovery. Thus, when implementing programs to improve the QOL of post COVID19 patients, the most affected domains of QOL and associated factors should be considered.

**Data Availability Statement:** All relevant data are within the paper and its Supporting information files.

**Funding:** The authors received no specific funding for this work.

**Competing interests:** The authors have declared that no competing interests exist.

## Background of the study

Coronavirus disease 2019 (COVID 19) is an infectious disease caused by novel coronavirus (SARS-CoV-2), a member of the family Coronaviridae [1]. Fever, cough, sore throat, dyspnea, myalgia or fatigue are all common clinical manifestations of the disease [2]. Several studies have discovered that some of these manifestations persist in patients even after being discharged or recovered from the disease [3–6]. Patients experience significantly higher levels of post-traumatic stress symptoms and depression due to the disease's novelty and the persistence of the symptoms [4, 7]. This has been severely affected to the patient's quality of life (QOL). Clinically stable COVID 19 patients can also be presented with depressive symptoms and lower QOL after the recovery [8]. It is important and timely to ascertain the impact of COVID 19 on those affected to assist healthcare professionals and government agencies in providing them with better support in advance. QOL is a widely used indicator for assessing and evaluating one's health and wellbeing. At least 150 different instruments are available to assess a person's QOL. Of them, SF-36, SF-12, EQ-5D-5L and EQ-5D-3L are most widely used in different settings throughout the world [9].

In this context, studies on the QOL of COVID 19 patients following discharge or recovery has been grown very rapidly. Therefore, it is necessary and timely to compile global evidence on the QOL of COVID 19 patients following discharge or recovery. In light of this importance, this systematic review was conducted to pool the scientific evidence available on QOL and its factors associated with it among adult survivors of COVID 19, at different timelines.

## Methods

This systematic review followed the updated guidelines of Preferred Reporting Items for Systematic Reviews and Meta-Analyses (PRISMA) and the protocol of the study was registered with the International Prospective Register of Systematic Reviews (CRD42021262639) on 30.07.2021.

### Eligibility criteria

Studies were included in the review if they (1) assessed the QOL among COVID 19 patients after the discharge or recovery, (2) were written in English, (3) used a validated instrument, and (4) used an observational or cohort study design. The studies that assessed the effectiveness of an intervention on QOL, were limited to children or people under the age 18 years of age and reported only the change in overall QOL rather than the various aspects, were excluded from this review. Besides, short abstracts, conference papers, reviews, viewpoints, short communications and preprints were excluded.

### Search strategy

Databases of PubMed, Cochrane Library, and Science Direct were searched from 01.11. 2019 to 31.06. 2021 using the combinations of keywords for QOL and COVID-19. The basic search strategy was built based on the research question formulation (PICO) and the search was done without restricting to a specific location to capture all the relevant titles. In order to identify the additional studies, a manual search was performed to identify the relevant studies included in reference lists of included studies and reviews. Medical subject headings (MeSH) were used where appropriate. Search terms were COVID 19 patients AND ("quality of life" OR "value of life" OR "quality of wellbeing" OR HRQOL OR "health related quality of life" OR "health quality of life" OR SF12 OR SF36 OR QOLIE).

## Selection process and screening

The studies identified through the database search were first exported into EndNote X9 reference management software and the duplicates were removed through the same software. The processes of screening of the identified studies were done independently by two authors (HMRKGN and PTSP) at both stages of title and abstract screening and full-text screening using the Rayyan systematic review software. Any discrepancies at both stages were resolved by consensus or consulting a third reviewer.

## Data extraction

Two reviewers independently (HMRKGN and PTSP) extracted the data from the included full texts into an excel sheet and cross-checked to ensure accuracy. Extracted information included the publication details (publication year, authors, country), study details (study design, setting, tools used for data collection), characteristics of the studied sample (age, sex, and sample size), QOL after the hospital discharge or recovery (mean value of total QOL, most affected domain and least affected domain), and factors associated with the low levels of QOL.

## Assessment of the quality of the studies

Quality assessment of the included studies was done by two reviewers independently (HMRKGN and AMMPA), using the Quality Assessment Tool for Observational Cohort and Cross-Sectional Studies of the National Heart, Lung, and Blood Institute of the National Institutes of Health (NHLBI) [10]. Any disagreement was resolved by consensus or by consulting a third reviewer. The quality assessment tool contained fourteen items including (1) Was the research question or objective in this paper clearly stated? (2) Was the study population clearly specified and defined? (3) Was the participation rate of eligible persons at least 50%? (4) Were all the subjects selected or recruited from the same or similar populations (including the same time period)? Were inclusion and exclusion criteria for being in the study prespecified and applied uniformly to all participants? (5) Was a sample size justification, power description, or variance and effect estimates provided? (6) For the analyses in this paper, were the exposure(s) of interest measured prior to the outcome(s) being measured? (7) Was the timeframe sufficient so that one could reasonably expect to see an association between exposure and outcome if it existed? (8) For exposures that can vary in amount or level, did the study examine different levels of the exposure as related to the outcome (e.g., categories of exposure, or exposure measured as continuous variable)? (9) Were the exposure measures (independent variables) clearly defined, valid, reliable, and implemented consistently across all study participants? (10) Was the exposure(s) assessed more than once over time? (11) Were the outcome measures (dependent variables) clearly defined, valid, reliable, and implemented consistently across all study participants? (12) Were the outcome assessors blinded to the exposure status of participants? (13) Was loss to follow-up after baseline 20% or less? (14) Were key potential confounding variables measured and adjusted statistically for their impact on the relationship between exposure(s) and outcome(s)?

## Data analysis

Data analysis was performed after including and excluding studies based on the quality assessment. Each study was evaluated descriptively and presented in tabular form.

## Ethical considerations

This study did not require ethical approval because this review was based on publicly available scientific literature.

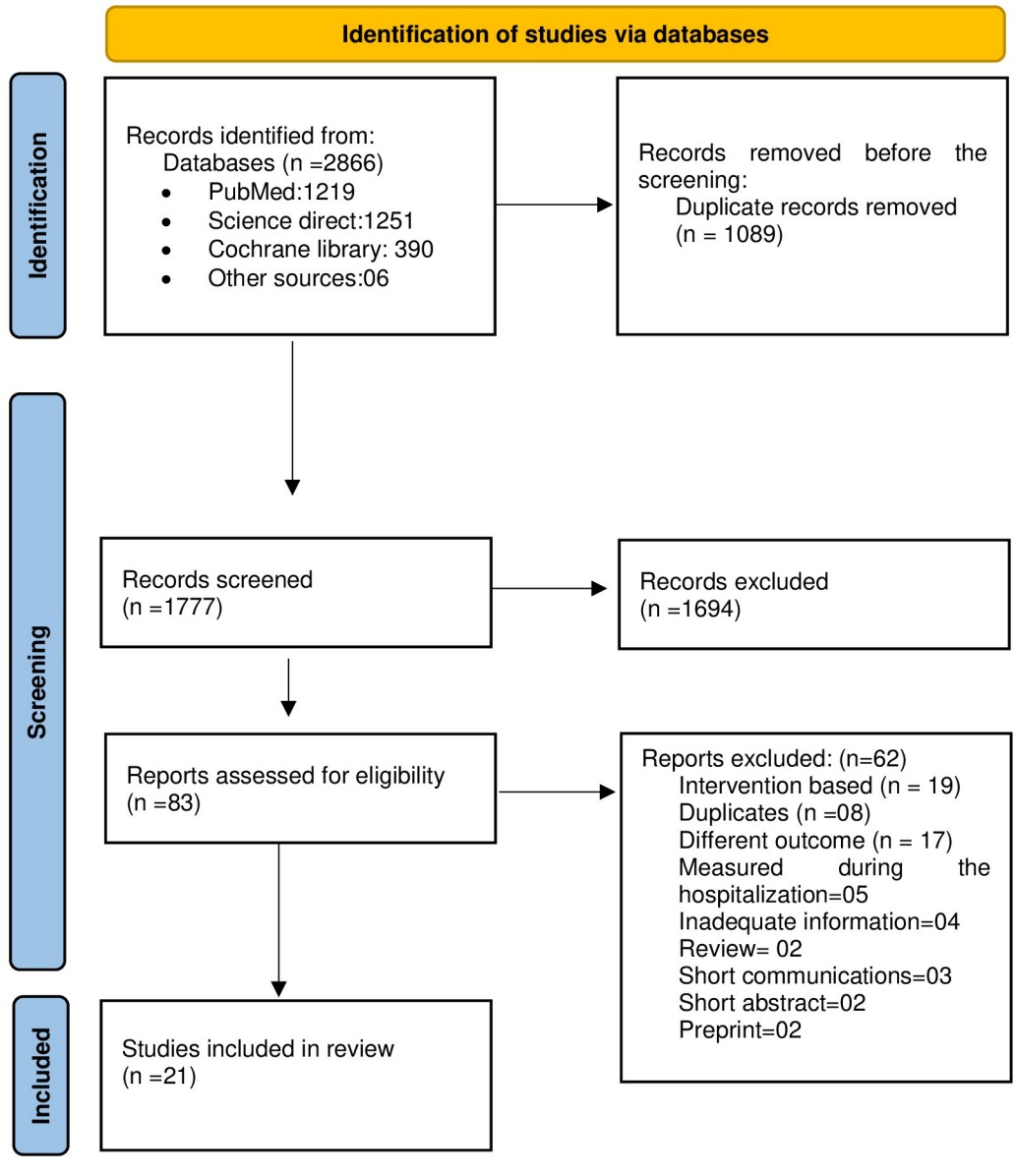

**Fig 1. Study selection.**

## Results

A total of 2866 studies were identified through the database search. After the duplicates were removed, there were 1777 studies for the title and abstract screening. 1694 studies were excluded during the title and abstract screening, and 83 studies were screened at the full-text screening stage. The final systematic review included only 21 studies [11–31] (Fig 1) and reasons for excluding full-texts were noted at the full-text screening stage.

### Quality assessment of the studies

Out of the 14 criteria used to assess the quality of study three criteria were met by all the studies. The research question/objective, inclusion/ exclusion criteria, study population and the

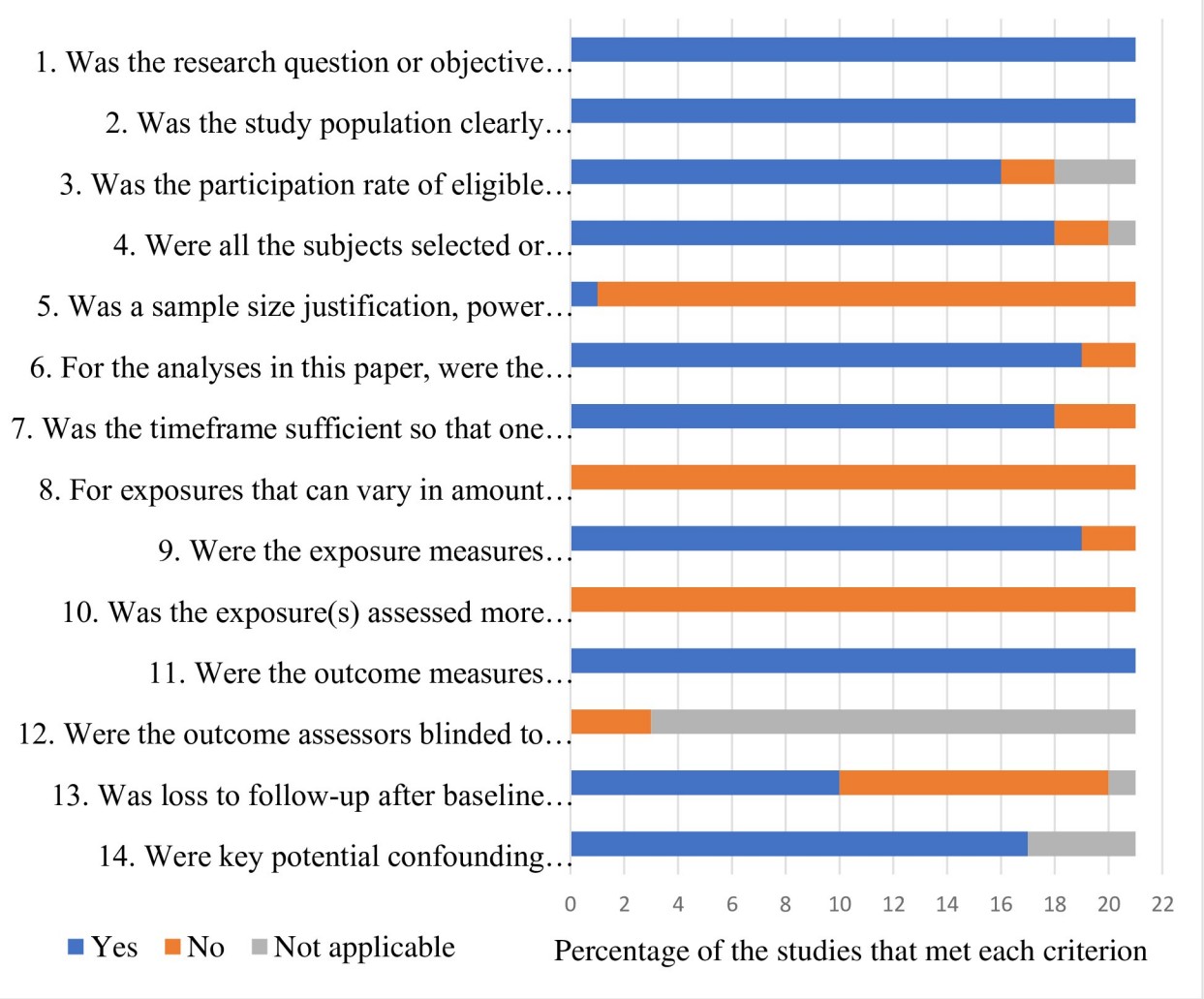

**Fig 2. Quality assessment of the studies.**

outcome measures were all clearly stated in all of the studies included in this review. The participation rates and the follow-up rates of the included studies were adequate. All studies were carried out during the pandemic period and outcomes were measured among COVID 19 patients after hospital discharge or recovery (Fig 2).

### Characteristics of the included studies

The sample sizes of the included studies ranged from 18 to 735, including a total of 4408 COVID 19 positive patients. Of the total number of patients, 50.2% (n = 2212) were males. Except for one study, all the other studies reported the mean age of the patients, and the age ranged from 41 to 74 years. Out of the total studies included, six studies [12, 14, 16, 21, 23, 28] were from Italy, three [13, 22, 31] from China, two from Spain [19, 27] and Norway [17, 30], and one each from Brazil [29], Lithuania [26], Nederland [20], Germany [15], France [18], Sweden [24], and Iran [11]. Another study [25] has been conducted as a global online survey.

In the included studies, QOL was assessed using different tools, including the SF-36, SF-12, EQ-5D-5L, EQ-5D-3L, 15-D instrument and St George's Respiratory questionnaire. The tools

**Table 1. Characteristics of the included studies.**

| No. | Author, Year | Country | Type of study | Participants | | | | QOL | |
|-----|--------------|---------|---------------|-------------|----------|------|--------|--------------------|----------------------|
| | | | | Sample size | Mean age | Male | Female | Measurement method | Time of assessment† |
| 1. | Wu et al., 2021 | China | Cross sectional | 27 | 63 | 8 | 19 | SF-36 | 6 months |
| 2. | Santus et al., 2020 | Italy | Cross sectional | 20 | 55 | 17 | 3 | St George's Respiratory questionnaire | 15 days |
| 3. | Temperoni et al., 2021 | Italy | Cross sectional | 104 | 41 | 56 | 48 | SF-36 | 1 month |
| 4. | Todt et al., 2021 | Brazil | Cohort study | 251 | 54 | 150 | 101 | EQ-5D-3L | 3 months |
| 5. | Strumiliene et al., 2021 | Lithuania | Cohort study | 51 | 56 | 25 | 26 | SF-36 | 2 months |
| 6. | Meys et al., 2020 | Nederland | Cross sectional | 210 | 45 | 26 | 184 | EQ-5D-5L | 79 +-17 days |
| 7. | Arab-Zozani et al., 2020 | Iran | Cross sectional | 409 | 58 | 247 | 162 | EQ-5D-5L | 14 days to 30 days |
| 8. | Walle-Hansen et al., 2021 | Norway | Cohort study | 106 | 74 | 60 | 46 | | 6 months |
| 9. | Qu et al., 2021 | China | Cohort study | 540 | NR | 270 | 270 | SF-36 | 3 months |
| 10. | Schandl et al., 2021 | Sweden | Cohort study | 113 | 58 | 86 | 27 | SF-36 | Between 2 to 7 months |
| 11. | Shah et al., 2021 | Multi-country | Cross sectional | 735 | 48 | 172 | 563 | EQ-5D-3L | 12.76 mean days (6.104) |
| 12. | Cinel et al., 2021 | Italy | Cohort study | 251 | 62 | 179 | 72 | EQ-5D-5L | 1 month and 3months |
| 13. | Garrigues et al., 2020 | France | Cross sectional | 120 | 63 | 75 | 45 | EQ-5D-5L | 100 days |
| 14. | Chen et al., 2020 | China | Cohort study | 361 | 47 | 186 | 175 | SF-36 | 1 month |
| 15. | Garratt et al., 2021 | Norway | Cross sectional | 458 | 50 | 202 | 256 | EQ-5D-5L | 4 months |
| 16. | Taboada et al., 2021 | Spain | Cross sectional | 91 | 66 | 59 | 32 | EQ-5D-3L | 6 months |
| 17. | Gamberini et al., 2021 | Italy | Cross sectional | 278 | 65 | 206 | 72 | 15-D instrument | 90 days |
| 18. | Carenzo et al., 2021 | Italy | Cohort study | 47 | 59 | 37 | 10 | EQ-5D-5L | 2 months and 6 months |
| 19. | Mendez et al., 2021 | Spain | Cross sectional | 179 | 57 | 105 | 74 | SF-12 | 2 months |
| 20. | Daher et al., 2021 | Germany | prospective study | 18 | 61 | 11 | 7 | EQ-5D-5L | 6 months |
| 21. | Monti et al., 2021 | Italy | Cross sectional | 39 | 56 | 35 | 4 | EQ-5D-3L | 2 months |

†After discharging from the hospital or after recovery.

of SF-36 [13, 17, 22, 24, 26, 28, 31] and EQ-5D-5L [11, 12, 14, 15, 18, 20, 30] were used by seven studies, while EQ-5D-3L was used by four studies [21, 25, 27, 29], SF-12 [19], and 15-D instrument [16] and St George's Respiratory questionnaire [23] were used by one study. The assessment of QOL in the included studies was done at different stages following hospital discharge or recovery and it ranged from 15 days to six months. Four studies assessed the QOL of discharged or recovered patients at the third [16, 18, 22, 29] and the sixth months [15, 27, 30, 31] respectively. Two studies assessed the QOL in the same group at two different times [12, 14]. Table 1 shows the characteristics of the included studies.

## Quality of life

As the QOL was assessed using different tools in different studies, the results of this review are explained based on the type of QOL assessment tool.

Out of the 21 studies, seven studies used SF-36 which is a very popular scale to measure QOL all over the world. It has eight subscales as physical functioning (PF), role physical (RP), bodily pain (BP), general health (GH), vitality (VT), social functioning (SF), role emotional (RE), and mental health (MH). Out of the seven studies that used SF-36, six reported the mean

**Table 2. Mean scores of QOL reported in the included studies.**

| Study | Measurement method | QOL total (Mean) | Most affected domain | Least affected domain |
|---|---|---|---|---|
| Chen et al., 2020 | SF-36 | 79.96 | Social function | Physical function |
| Temperoni et al., 2021 | SF-36 | 52.7 | Physical role | Physical function |
| Strumiliene et al., 2021 | SF-36 | NR | Physical role | Physical function |
| Schandl et al., 2021 | SF-36 | 69 | Physical role | Mental Health |
| Qu et al., 2021 | SF-36 | 64.72 | Mental Health | Physical function |
| Wu et al., 2021 | SF-36 | 67.22 | Reported heath transition | Role emotional |
| Garratt et al., 2021 | SF-36 | 74.07 | Energy/Fatigue | Physical function |
| Arab-Zozani et al., 2020 | EQ-5D-5L | 0.61 | Mobility | Self-care |
| Daher et al., 2021 | EQ-5D-5L | 1.8 | Mobility | All other areas |
| Garrigues et al., 2020 | EQ-5D-5L | 0.86 | Pain/discomfort | Self-care |
| Cinel et al., 2021 | EQ-5D-5L | 1.26 at one month 1.28 at three months | Pain/discomfort | Self-care |
| Meys et al., 2020 | EQ-5D-5L | 0.62 | Usual activities | Self-care |
| Carenzo et al., 2021 | EQ-5D-5L | NR | 2 months: Usual activities 6 months: Pain/discomfort | Self-care |
| Walle-Hansen et al., 2021 | EQ-5D-5L | < 75 years = 1.67 >75 years = 1.94 | <75 years: Pain/discomfort > 75 years: Usual activities | <75 years: Self-care >75 years: Self-care |
| Taboada et al., 2021 | EQ-5D-3L | NR | Pain/discomfort | Self-care |
| Monti et al., 2021 | EQ-5D-3L | NR | Pain / discomfort | Self-care |
| Todt et al., 2021 | EQ-5D-3L | NR | Pain / discomfort | Self-care |
| Shah et al., 2021 | EQ-5D-3L | 8.65 | Usual activities | Self-care |
| Mendez et al., 2021 | SF-12 | 44 | Physical component | Mental component |
| Santus et al., 2020 | St George's Respiratory questionnaire | NR | Activity change | Social and emotional impact |
| Gamberini et al., 2021 | 15-D instrument | 0.850 | Sexual activities | Hearing |

value for the total scores, ranging from 52.7 to 79.9. Of those, three studies reported the physical role as the least scored subscale of the SF-36, which means it is the most affected domain of the QOL. Five studies reported physical function as the highest scored subscale which means the least affected domain of QOL.

Seven other studies reported QOL using the EQ-5D-5L scale, a valid and reliable scale with five subdimensions such as mobility, self-care, usual activities, pain/discomfort, and anxiety/depression. Each dimension has five levels: no problems, slight problems, moderate problems, severe problems, and extreme problems. Similar to EQ-5D-5L, EQ-5D-3L also has the same dimensions and each dimension has three levels: no problems, some problems, and extreme problems. In the present review, four studies used EQ-5D-3L scale to access the QOL. Therefore, altogether 11 studies out of 21 used EQ-5D scales. Out of the seven studies that used EQ-5D-5L, six reported a mean value for the total score ranging from 0.61 to 1.94. Nevertheless, the highest mean of the total score (1.94) was reported by a study done among patients aged >75 years. According to all the studies (11 studies) which used the EQ-5D scale, all studies identified the self-care domain as the domain with the least affected domain of QOL. Seven studies identified the pain/discomfort domain as the most affected domain of QOL. Other commonly affected domains were mobility and usual activities. Table 2 shows the details of QOL according to the different tools used.

## Factors associated with the low level of quality of life

Out of the 21 studies included, ten studies identified factors associated with the low levels of QOL among discharged or recovered COVID 19 patients. Seven studies [11, 13, 16, 25, 27, 29,

**Table 3. Factors associated with the low levels of QOL.**

| Study | Associated factors |
|---|---|
| Todt et al., 2021 | Female sex, older age, Having diabetes and hypertension, Heart failure, Number of comorbidities, Highest respiratory support required, Intensive care admission, New onset hemodialysis, Longer duration of hospital stay, |
| Strumiliene et al., 2021 | Reduced lung function |
| Arab-Zozani et al., 2020 | Female sex, older age, Higher education level, Being unemployed, ICU admission, Having diabetes |
| Walle-Hansen et al., 2021 | Older age |
| Qu et al., 2021 | Female sex, older age, Poor hemoglobin and albumin level |
| Shah et al., 2021 | Female sex, Hospitalized survivors, Higher number of weeks since COVID 19 diagnosis |
| Chen et al., 2020 | Female sex, Older age, Severity of the clinical subtype, Higher length of hospital stay |
| Taboada et al., 2021 | Older age, Male sex, Use of mechanical ventilation during ICU stay, Higher duration of mechanical ventilation, Longer duration of ICU stay, Longer duration of hospital stay |
| Gamberini et al., 2021 | Female sex, Older age, Prolonged mechanical ventilation, Present with comorbidities |
| Mendez et al., 2021 | Neurocognitive impairment, Positive screening for psychiatric morbidity |

30] identified that older age is associated with low levels of QOL while six studies [11, 13, 16, 22, 25, 29] identified that low levels of QOL are associated with the female sex. Admission to an ICU, prolonged mechanical ventilation or longer ICU stay are associated with a low level of QOL as reported by four studies [11, 16, 27, 29]. Furthermore, three studies [11, 16, 29] reported that the patients with other co-morbidities had lower levels of QOL than patients without any co-morbidity. Table 3 shows the factors associated with the low level of QOL.

## Discussion

This systematic review presents the pooled results of published literature on QOL among COVID19 patients following the hospital discharge or recovery. Twenty-one studies met the inclusion criteria and of those, 12 studies were cross sectional, and nine were cohort studies. The present review included articles published from the beginning of the pandemic in December 2019 to July 2021. Although COVID 19 is a novel disease, there was a good representation of studies that assessed the QOL among COVID 19 patients with different characteristics at different time frames.

The most common factors associated with a low level of QOL are female sex, older age, the presence of co-morbidities, ICU admission, prolonged ICU stay, and mechanical ventilation. Regardless of the time since discharge or recovery, QOL has been greatly affected. According to the studies done using SF-36, the most affected domain and least affected domain of the QOL were physical role and physical function, respectively. The studies which utilized the EQ-5D scale revealed that pain/discomfort was the most affected domain and self-care was the least affected domain of QOL. As the QOL had been measured using different tools in these studies, it was difficult to understand the most affected domain of the QOL considering the results of all the studies included in this review. Nevertheless, studies have found an excellent criterion validity between the EQ-5D subscales and SF-36 subscales [32]. A study done by Rowen et al., also suggested that models mapping the SF-36 onto the EQ-5D have similar predictions across inpatient and outpatient setting and medical conditions [33]. According to a study done by Hawthorne et al., the 15D instrument also demonstrated a good correlation with SF-36 [34]. Since most of the studies (18 studies) included in this review used SF-36 and

EQ-5D tools to assess QOL, the reported changes of QOL of discharged or recovered COVID-19 patients might be collated.

The most common factors associated with a low level of QOL are female sex, older age, the presence of co-morbidities, ICU admission, prolonged ICU stay, and mechanical ventilation. Most of the studies available in the literature to assess the sex differences in QOL have found that QOL is significantly lower among females than males [35, 36] and women's mental health has been disproportionately impacted by the COVID 19 pandemic, even if they have not had COVID-19 [37]. Contrary to the findings reported by all these studies, only one study included in this review found that females had better QOL than males.

Patients developing critical illness can have substantial attributable mortality rates and it may cause drastic changes in their QOL. Similar reviews conducted among patients after ICU discharge also found that the ICU survivors had significantly lower mean scores of QOL during the post-discharge follow-up [23, 38–40]. Therefore, it is clearly affirmed that the COVID 19 patients with ICU admission and longer duration of admission also reported low QOL after discharge compared to others.

## Strengths and limitations

The key strength of this systematic review is that it was timely conducted 1.5 years following the pandemic, when many countries have conducted their studies to assess the QOL among COVID 19 patients after discharge or recovery. Though it was a short period, we found enough studies on QOL among COVID 19 patients. By including a variety of studies representing 11 countries in the world and COVID 19 patients with different characteristics and different time frames after discharge or recovery, the findings of this review can be generalized to explicit the QOL among COVID 19 patients after discharge or recovery. Furthermore, this review was conducted according to the international guidelines after obtaining the registration for the protocol in PROSPERO.

There were several limitations. Firstly, the included studies in this review assessed the QOL using different tools, making it difficult to make a fair comparison. For the same reason, QOL was not presented quantitatively as a whole. Secondly, limiting the language to English may result in the omission of important findings reported in other languages. Finally, as the article search was done using only three databases, some relevant articles might have been missed. Considering all the limitations, the findings of this systematic review should be interpreted with caution.

## Conclusion

QOL of the post COVID-19 patients was greatly affected, regardless of the time elapsed since discharge or recovery. Policymakers and healthcare providers must urgently investigate robust strategies for improving the QOL of post COVID-19 patients. Patients of older age, those with co-morbidities, those admitted to the ICU, those who stayed in ICU for a long time, and those who were mechanically ventilated have a higher risk of poor QOL following the infection. Healthcare providers should take extra precautions to improve their quality of life after the infection. Following recovery, women in particular, should receive additional attention because they are more likely to have poor QOL. Community-based healthcare programmes, at least via virtual forms, are recommended to improve the QOL of post COVID-19 patients and the most affected domains of QOL and the associated factors should be considered while implementing programmes.

## Supporting information

**S1 Checklist. PRISMA checklist.**
(DOCX)

**S1 Data. Data extraction sheet.**
(XLSX)

## Author Contributions

**Conceptualization:** H. M. R. K. G. Nandasena, P. T. S. Prasanga.

**Formal analysis:** H. M. R. K. G. Nandasena, M. L. Pathirathna, A. M. M. P. Atapattu, P. T. S. Prasanga.

**Methodology:** H. M. R. K. G. Nandasena, M. L. Pathirathna, A. M. M. P. Atapattu, P. T. S. Prasanga.

**Project administration:** H. M. R. K. G. Nandasena.

**Supervision:** H. M. R. K. G. Nandasena.

**Writing – original draft:** H. M. R. K. G. Nandasena.

**Writing – review & editing:** H. M. R. K. G. Nandasena, M. L. Pathirathna, A. M. M. P. Atapattu, P. T. S. Prasanga.

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
