## [Decision Letter · Decision Letter 0]

24 Jan 2022

PONE-D-21-35133Quality of life of COVID 19 patients after discharge: Systematic reviewPLOS ONE

Dear Dr. Nandsena,

Thank you for submitting your manuscript to PLOS ONE. After careful consideration, we feel that it has merit but does not fully meet PLOS ONE’s publication criteria as it currently stands. Therefore, we invite you to submit a revised version of the manuscript that addresses the points raised during the review process.

We look forward to receiving your revised manuscript.

Kind regards,

Mohamed A Yassin, MD

Academic Editor

PLOS ONE

Journal Requirements:

Reviewers' comments:

Reviewer's Responses to Questions

**Comments to the Author**

1. Is the manuscript technically sound, and do the data support the conclusions?

Reviewer #1: Yes

Reviewer #2: Yes

2. Has the statistical analysis been performed appropriately and rigorously? 

Reviewer #1: Yes

Reviewer #2: Yes

3. Have the authors made all data underlying the findings in their manuscript fully available?

Reviewer #1: Yes

Reviewer #2: Yes

4. Is the manuscript presented in an intelligible fashion and written in standard English?

Reviewer #1: Yes

Reviewer #2: Yes

5. Review Comments to the Author

Reviewer #1: I would first like to extend my admiration of their work to the authors.

It is indeed a greatly written manuscript and analyzed work.

-Some fine tuning of the contents is needed to strictly match the requirements of PLOS manuscript guidelines.

-The one reservation I have, is the issue of ethical consideration. Though rarely discussed in the context of systematic reviews, I believe the authors should address the matter as it is associated with potential conflicts of interest and issues of data representation.

-Additionally, it would be better to expand the conclusion segment to highlight how your review impacts current public health practice.

Reviewer #2: This is a well-written paper, providing insight into the quality of life post-COVID 19. However, the following concerns will need to be addressed.

1. Grammatical errors/ incomplete sentences; ¶& both the pilcrow and ampersand symbols represent equal contribution to the work, you could stick to one. Will the use of female sex be more suitable compared with female gender? In the background; Clinically stable COVID 19 patients can also be presented with depressive symptoms and lower quality of life following recovery (doesn't read well). There are at least 150 instruments exist to assess a person's QOL...(check sentence). In this context, the research evidence on the QOL of COVID19 patients following discharge or recovery has grown at very rapidly. In the Quality assessment of the studies section; Out of 14 criteria used to assess the quality of a study were met by all the studies. (check wording). 2. Is there a numerical correlation between the EQ-5D subscales and the SF-36 subscales that you could include in the discussion?

6. PLOS authors have the option to publish the peer review history of their article (what does this mean?). If published, this will include your full peer review and any attached files.

Reviewer #1: No

Reviewer #2: No

---

## [Author Response · Author response to Decision Letter 0]

27 Jan 2022

Thank you for giving the opportunity to revise the previous version of our manuscript. We appreciate the careful review and constructive suggestions given by the reviewers. We believe that the manuscript is substantially improved after making the suggested revisions. Our responses are given in a point-by-point manner including how and where the text was modified. Changes made in the manuscript are highlighted in the revised document.

---

## [Editor Report · Decision Letter 1]

31 Jan 2022

Quality of life of COVID 19 patients after discharge: Systematic review

PONE-D-21-35133R1

Dear Dr. Nandasena,

We’re pleased to inform you that your manuscript has been judged scientifically suitable for publication and will be formally accepted for publication once it meets all outstanding technical requirements.

Kind regards,

Mohamed A Yassin, MD

Academic Editor

PLOS ONE

Additional Editor Comments (optional): The manuscript  is accepted for publication in its current  form
---

## [Editor Report · Acceptance letter]

4 Feb 2022

PONE-D-21-35133R1 

Quality of life of COVID 19 patients after discharge: Systematic review 

Dear Dr. Nandasena:

I'm pleased to inform you that your manuscript has been deemed suitable for publication in PLOS ONE. Congratulations! Your manuscript is now with our production department. 

Kind regards, 

on behalf of

Dr. Mohamed A Yassin 

Academic Editor

PLOS ONE